# Viral Pathogenesis, Recombinant Vaccines, and Oncolytic Virotherapy: Applications of the Canine Distemper Virus Reverse Genetics System

**DOI:** 10.3390/v12030339

**Published:** 2020-03-20

**Authors:** Jianjun Zhao, Yanrong Ren, Jie Chen, Jiasan Zheng, Dongbo Sun

**Affiliations:** 1College of Animal Science and Veterinary Medicine, Heilongjiang Bayi Agricultural University, Daqing 163319, Chinadongbosun@126.com (D.S.); 2Institute of Special Animal and Plant Sciences, Chinese Academy of Agricultural Sciences, Changchun 130112, China; renyanrong0927@163.com (Y.R.); chenjie@caas.cn (J.C.)

**Keywords:** canine distemper virus, reverse genetic system, pathogenesis, recombinant vaccine, virotherapy

## Abstract

Canine distemper virus (CDV) is a highly contagious pathogen transmissible to a broad range of terrestrial and aquatic carnivores. Despite the availability of attenuated vaccines against CDV, the virus remains responsible for outbreaks of canine distemper (CD) with significant morbidity and mortality in domesticated and wild carnivores worldwide. CDV uses the signaling lymphocytic activation molecule (SLAM, or CD150) and nectin-4 (PVRL4) as entry receptors, well-known tumor-associated markers for several lymphadenomas and adenocarcinomas, which are also responsible for the lysis of tumor cells and apparent tumor regression. Thus, CDV vaccine strains have emerged as a promising platform of oncolytic viruses for use in animal cancer therapy. Recent advances have revealed that use of the CDV reverse genetic system (RGS) has helped increase the understanding of viral pathogenesis and explore the development of recombinant CDV vaccines. In addition, genetic engineering of CDV based on RGS approaches also has the potential of enhancing oncolytic activity and selectively targeting tumors. Here, we reviewed the host tropism and pathogenesis of CDV, and current development of recombinant CDV-based vaccines as well as their use as oncolytic viruses against cancers.

## 1. Introduction

Canine distemper virus (CDV) is an enveloped negative-RNA virus, which, along with measles virus (MeV), Rinderpest virus (RPV), peste des petits ruminants virus (PPRV), phocine distemper virus (PDV), Cetacean morbillivirus (CeMV), and recently discovered feline morbillivirus (FeMV), belongs to the *Morbilivirus* genus within the *Paramyxoviridae* family [1,2]. Noted, CDV is the etiological agent of canine distemper (CD) that has been known since the mid-1700s and might have originated from the infection of dogs by MeV during human epidemics in the New World [3]. Both viral agents are known to be highly contagious, spread via the respiratory route, and cause a similar pathogenesis characterized by fever, skin rash, and conjunctivitis with profound immune suppression, but also elicit lifelong immunity in surviving hosts [4,5,6,7]. However, MeV has a narrow natural host range restricted to humans and certain non-human primates, whereas CDV can infect most of the terrestrial and aquatic carnivorous species, even non-human primates. Over the past decades, CDV has caused several fatal outbreaks in wild carnivores and non-human primates, including the deaths of almost all African wild dogs (*Lycaon pictus*), about 10,000 Caspian seals (*Phoca caspica*), around 35% of Serengeti lions (*Panthera leo*), and 30% of rhesus monkeys (*Macaca mulatta*) [8,9,10].

Currently, attenuated CDV strains are routinely used worldwide as live vaccines against CDV infection in dogs, minks, and other carnivores. However, these vaccines have often been reported as insufficiently attenuated for ferrets and other wildlife species and can cause symptomatic and sometimes fatal infections after administration, resulting in an increased interest for a safe and efficient CDV vaccine for highly susceptible species [11,12,13,14,15]. In addition, CDV can induce apoptosis in many cancer cell lines, including canine neoplastic lymphocytes, canine adenofibrosarcoma cells, human breast cancer cells, and human cervical cancer cells. Moreover, persistent infection by CDV has been shown to exhibit anti-angiogenic properties in a canine histiocytic cell line [16,17,18,19,20]. These findings suggest that CDV could be used as an oncolytic virus in oncolytic gene therapy in both humans and animals.

This review highlights the potential use of the CDV genome as a viral vector for the development of novel vaccines and oncolytic virotherapy against cancers. Specifically, the development of a CDV reverse genetics system (RGS) and identification of the viral cellular receptors over the past decades, contributed to the understanding of viral pathogenesis and the generation of recombinant CDV-based vaccines, as well as the exploitation of the oncolytic potential of the virus against cancers by targeting tumor markers as viral receptors.

## 2. CDV Reverse Genetic System (RGS)

The development of the RGS of RNA viruses can be used to produce modified viruses that have specific properties with potential applications in vaccine development, gene therapy, and basic science. Concomitantly, foreign genes were inserted to (1) allow localization of viral replication in vivo through the expression of marker genes, (2) develop candidate multivalent vaccines against vector viruses and other pathogens, and (3) create candidate oncolytic viruses. The vector use of RNA viruses was experimentally encouraged by the pronounced and unexpected genetic stability of the recombinants, and by the high load of insertable genetic material [21].

### 2.1. CDV Genome Organization

The CDV genome is 15,690 nucleotides in length and consists of a short 3′ leader region and 6 genes encoding the nucleocapsid (N), phospho (P), matrix (M), fusion (F), hemagglutinin (H), and large (L) proteins, which are separated by intergenic regions of 3 nucleotides and followed by a short 5′ trailer region. The V and C nonstructural proteins are encoded within the P gene. The V protein is expressed by cotranscriptional RNA editing, and C is expressed from an overlapping reading frame [22]. The N, P, and L proteins together with the viral RNA constitute the ribonucleoprotein (RNP) complex, which either directs the sequential synthesis of capped and polyadenylated mRNAs from 6 transcription units or the replication of full-length encapsidated antigenomes. The H glycoprotein mediates the binding of the virus to the cell membrane, and the F glycoprotein executes the fusion of the 2 membranes, enabling the entry of the viral RNP into the cytoplasm and the initiation of an infectious cycle [23,24]. As CDV is a member of the morbilliviruses, it is known that its genomic or antigenomic RNA, as for all *Mononegavirales*, cannot function as mRNA. Thus, to generate the viral RNP, the minimal infectious unit of negative-strand RNA viruses, simultaneous expression of viral protein components forming the polymerase complex, and full-length viral antigenomic RNA (cRNA) from cDNA are required [1,21].

### 2.2. Development of CDV RGS from Vaccine to Wild-Type Strains

In standard RNA virus RGSs, an exact 3′-end of the antigenome is generated by autocatalytic cleavage from a hepatitis delta virus ribozyme (HDV-Rbz) sequence positioned directly downstream of the antigenome. Accordingly, the first established CDV RGS was the Vero cell-adapted Onderstepoort vaccine strain, which was recovered from Vero cells overlaying vaccinia-T7 virus-infected HeLa or 293 cells after cotransfection with a plasmid encoding the full-length cDNA (driven by an optimal T7-promoter comprising three extra guanine (G) nucleotides at the 3′-end), and supporting plasmids encoding the N, P, and L proteins [25,26]. The rescued virus revealed no major differences with the parental virus in the growth kinetics or in the shape and size of syncytia in Vero cells. Although the use of the optimal T7 promoter enhances the processivity and transcript levels of the T7 RNA polymerase [27], in the context of paramyxovirus RGS, the optimal T7 promoter has mixed effects on CDV rescue because these extra three G nucleotides not only interfere with appropriate recognition of the genomic terminus by the RdRp complex, but also interfere with the “rule of six”, a property shared by all members of the *Paramyxoviridae* family. The T7 RNA polymerase in this RGS was delivered by MVA-T7, a derivative of the modified vaccinia Ankara (MVA) virus, which can drive high-level expression of T7 RNA polymerase in the cytoplasm. However, the use of a vaccine virus has several disadvantages, primarily high poxvirus-mediated cytotoxicity, challenges associated with separating vaccinia virus from the rescued viruses, and increasing restrictions on the use of vaccinia virus, especially for vaccine development. To avoid these possible problems, cells lines stably expressing T7 RNA polymerase has been developed as alternatives to the use of helper T7-vaccinia virus. The most commonly used of which are BSR-T7 cells, a baby hamster kidney (BHK)-derived cell line stably expressing T7 RNA polymerase [28,29,30,31]. As T7 cell lines have significantly lower T7 expression than that of vaccinia virus-mediated systems [32], and as T7 expression has been shown to be a major determinant of virus rescue efficiency, this resulted in the low virus recovery efficiency of these RGSs. To overcome the low rescue efficiency that results from the use of an optimal T7 promoter and the low T7 expression levels in BSR-T7 cells, Beaty et al. developed a single-step transfection protocol [33]. By incorporating a self-cleaving hammerhead ribozyme (Hh-Rbz) sequence immediately before the 5′-end of the recombinant viral antigenome combined with the use of a codon-optimized T7 polymerase, the authors significantly improved T7 expression, thereby enabling a substantial increase in paramyxovirus rescue efficiency for representative viruses belonging to all five major *Paramyxoviridae* genera [33]. Instead of a T7-promoter-based RGS, alternatively, RNA polymerase II (Poll II)-controlled expression of antigenome and supporting expression plasmids were also used to improve the rescue efficacies of MeV and CDV from their cDNAs [34,35]; the lower precision of the start and stop of Poll II transcription may allow completion of the virus from genomes not in line with the “rule of six” [34].

Recently, several generated RGSs have been reported for the rescue of CDV wild-type strains from marmoset lymphoblastoid cells (B95a) or Vero cells expressing the signaling lymphocytic activation molecule (SLAM) receptor (Vero-SLAM) that allows for propagation of wild-type CDVs and maintenance of their virulence in vivo (Figure 1). Results have confirmed that CDV could be used as a vector for the expression of an enhanced green fluorescence protein (EGFP) in the form of an extra transcription unit without loss of its virulence, although the expression level of the foreign protein, as well as the replication efficiency and the virulence of the recombinant viruses depend on the site of the insertion of the EGFP gene [36,37,38]. Thus, RGS approaches for CDV have greatly increased the understanding of the pathogenesis of wild-type strains and the attenuating mechanisms of vaccines, as well as subsequently facilitated the generation of recombinant CDV-based vaccines. 

## 3. Understanding the Viral Pathogenesis Mechanisms through CDV RGS

### 3.1. Receptor-Dependent CDV Entry, Dissemination, and Shedding

Identification of cellular receptors has vastly improved our understanding of CDV viral tropism and pathogenesis. Two cellular receptors, the SLAM/CD150, and nectin cell adhesion molecule 4 (nectin-4/PVRL4), involved in the entry of morbillivirus in the cell, and subsequent infection, have been identified for both the vaccine and wild-type CDV strains. Noted, SLAM is reported to be expressed on the surface of activated T-lymphocytes, and B-lymphocytes, macrophages, and dendritic cells, but not in epithelial cells, whereas nectin-4 is known to be expressed on the apical and basolateral surface of polarized epithelial cells, as well as in endothelial cells in the central nervous system (CNS) of dogs [39,40,41,42]. Thus, both SLAM and nectin-4 have been suggested to play crucial roles in the entry, dissemination, and pathogenesis of CDV in vivo (Figure 2). After transmission by aerosol, CDV has been shown to cross the respiratory tract to reach immune tissues in the oral cavity via infected SLAM expressing immune cells, most likely macrophages or dendritic cells (Figure 2a). Then, infected cells are known to carry the virus to the draining lymph nodes, where the resident activated T-cells and B-cells are infected via SLAM, resulting in the amplification of the virus and the initiation of primary viremia. Subsequently, the virus gets disseminated to secondary lymphoid organs, including the spleen, thymus, lungs, and gastrointestinal tract, followed by a systemic spread through the entire immune circulation system, causing secondary viremia with severe immunosuppression (Figure 2b). At this point, CDV infection is thought to occur via the basolateral side of nectin-4 expressing lumen epithelium via the migration of virally-infected T-cells, B-cells, and dendritic cells from the circulation (Figure 2c). The hosts would show signs of CD clinical disease, including developing a rash, diarrhea, and conjunctivitis, only after CDV infection of epithelial cells. Then, CDV is known to shed from the epithelium of the respiratory and digestive route via its apical surface (Figure 2d). In later stages of infection, CDV would usually cross the blood–brain barrier and reach the CNS, causing neurological symptoms, which might be mediated by nectin-4 or a yet unknown cellular entry receptor in astrocytes [43,44,45,46,47].

The CDV RGS has also been used to identify the key residues in the H protein interacting with cellular receptors and evaluate their contribution to the viral entry. Recently, the P493S/Y539A double mutations in the H protein were introduced in an unable to recognize nectin-4 (nectin-4-blind) recombinant CDV strain, which was recovered and used to infect ferrets. Although the infected ferrets showed no clinical signs or viral shedding, there was a rapid and efficient spreading of immune cells observed, producing the same levels of leukopenia and inhibiting lymphocyte proliferation activities as in the case of wild-type strain infected animals. This finding highlights the fact that the SLAM-expressing immune cells appear to be necessary for CDV entry, dissemination, and initial infection, as well as immunosuppression, whereas infection of nectin-4 mediated epithelial cells seems to be required for viral shedding and clinical disease, as well as viral transmission [45]. Additionally, a very recent in vivo study in ferrets infected with SLAM- or nectin-4-blind CDV strains caused by mutations in key residues of the H protein confirmed that sequential use of the SLAM and nectin-4 receptors is essential for the transmission of CDV to naïve hosts [46]. A further study on three simultaneously administered recombinant wild-type CDVs expressing green (Venus), red (dTom), or blue (TagBFP) fluorescent proteins to ferrets, allowing assessment of viral entry, intra-host dissemination, and inter-host transmission, showed that CDV could efficiently enter the host if delivered to the nose or lung, as well as lead to infection of the host through conjunctival administration [48].

### 3.2. Virulence and Virally-Mediated Immunosuppression in Host

In addition to the identification of key residues in the H and F proteins sustaining the SLAM or nectin-4 receptor dependent cell entry and fusion in vitro [44,49,50], several CDV RGS studies were recently performed on the establishment of genetic mutations or gene knockout strains to study the mechanisms of virulence in vivo. These studies investigated the determinants of CDV virulence, including not only the envelope (H, F, and M) and replication proteins (N, P, and L), but also the accessory C and V proteins that counteract host defenses [30,43,51,52]. To understand how CDV might invade the host and cause immunosuppression, CDV variants, which were SLAM-blind or incapable of expressing either one or both of the nonstructural V and C proteins (C- or V-knockout, or both), were generated based on RGS approaches and used to infect ferrets, respectively. Results confirmed that the V protein was able to sustain the swift lymphocyte-based invasion of mucosal tissues and lymphatic organs, as well as the inhibition of the induction of interferon (IFN)-α/β in peripheral blood mononuclear cells (PBMCs) and that of other important cytokines, such as tumor necrosis factor α (TNF-α), IFN-γ, interleukin 6 (IL-6), and IL-4 that control the activation of cellular and humoral immune processes, whereas the C protein was shown to be dispensable for the invasion of the lymphatic organs and appeared to sustain subsequent infection phases [53]. Detailed molecular analysis of the neuropathogenic CDV A75/17 strain demonstrated that the V protein specifically ablates the nuclear import of the signal transducer and activator of transcription 1 (STAT1) and STAT2 without affecting their activated phosphorylation states [54]. A further study demonstrated that the V protein could block type I and II IFN responses through inhibition of the nuclear translocation of STAT2 and mda5 signaling, but not STAT1, which was observed to be critical for lethal disease in ferrets [55]. These results provided new insights into the mechanisms of immune evasion by CDVs and might lead to the development of new vaccines with reduced virulence and immunosuppressive properties.

## 4. History and Challenges Associated with CDV Vaccines

Both live-attenuated and inactivated CDV vaccines have been used to reduce the number of CD outbreaks in ferret, mink, and dog populations [56]. However, compared with live-attenuated vaccines, inactivated vaccines have been reported to provide insufficient protection due to a weaker inflammatory response to antigens [57]. Until recently, two live-attenuated vaccine strains against CDV, both introduced in the early 1960s, were available. The first live-attenuated vaccine strain, the Onderstepoort, was developed from a natural isolate from an infected fox, which was passaged in ferrets and then adapted to chicken embryos, and later replaced with chicken cell cultures [58]. The other was generated by adaptation of the Rockborn strain to canine kidney cell cultures [59]. The canine-adapted cell culture vaccine is known to offer protection to almost 100% of vaccinated dogs, but on rare occasions can cause post-vaccination encephalitis [12]. Although these two vaccines significantly reduced the number of CDV infections in domestic dog populations, they were insufficiently attenuated for use in wildlife species. For example, the Rockborn strain vaccine generated in canine cells was reported to cause disease in grey foxes (*Urocyon cinereoargenteus*) and ferrets (*Mustela nigripes*) [13,60], while the Onderstepoort strain vaccine might be fatal for both the European mink (*Mustela lutreola*) and ferrets [13,14]. Problems associated with live-attenuated vaccines, especially their unsuitability for many endangered species, have served as the incentive for the development of recombinant vaccines, which can be safely used across species, particularly for wildlife. Respectively, the recently generated recombinant CDV vaccine, incorporating the envelope proteins of CDV in a strain of canarypox virus, was shown to be safe to all susceptible species tested to date [61,62,63,64]. However, it is replication incompetent and as such induces a milder immunological response than live-attenuated vaccines, thus requiring regular reimmunization [65]. Recently, CDV DNA vaccines encoding viral nucleocapsid and envelope proteins have also shown promise by inducing a strong protective humoral and cell-mediated immune response and protecting dogs and minks against wild-type CDV challenges [66,67,68]. Although effective and relatively inexpensive to produce, problems associated with the vaccination regime and efficient delivery routes must be resolved to make DNA vaccines an effective alternative to live-attenuated vaccines for wildlife. 

## 5. CDV Recombinants as Safe and Efficient Candidate Multivalent Vaccines

Due to the availability of an RGS for manipulating CDV genomes and generating respective recombinant viruses, a number of recombinant viruses based on attenuated CDV strains has been generated that present antigens of foreign pathogens during CDV replication. These recombinant vaccines have an excellent safety record in animal models and elicit humoral as well as cellular immune responses in vaccinated animals, which are responsible for long-lasting protection (Table 1). 

CDV vaccine strains have recently been used as a vector to stably express the glycoprotein (G) of the rabies virus (RABV) within an additional transcription unit of a CDV cDNA clone. The RABV-G can be incorporated in the viral envelope of the chimeric virus to improve immunological recognition. The chimeric virus was shown to be safe for both mice and dogs and resulted in the production of long-lasting neutralizing antibodies against both CDV and rabies, and even protected mice from a lethal dose of rabies, demonstrating potential for the production of multivalent vaccines against both pathogens [31,35]. A recombinant avirulent CDV strain expressing *Leishmania* antigens (leishmania homolog of receptors for activated C-kinase (LACK), thiol-specific antioxidant (TSA), or protein antigen LmSTI1 (LmSTI1), was also reported suitable for use as a polyvalent vaccine vector for *Leishmania*, providing protection against both CDV and major infections of *Leishmania* in dogs [69]. In addition, the generation of recombinant CDV expressing interleukin (IL)-18 or IL-7 also showed that they could be potentially used as molecular immunoadjuvants and agents for anticancer therapies in vivo [70,71]. These results also highlighted the requirement for evaluation of feasible intergenic regions for the insertion of a foreign gene because it was shown that the polar gradient transcription of the CDV genome could constitute a limiting factor for the expression of a foreign gene. The further upstream the additional transcription unit (ATU) cassette, the higher the amount of mRNA transcribed and higher the protein expression. However, if the encoded foreign gene product interferes with CDV replication, high levels of the encoded antigen may be detrimental [71]. Another approach for generating a chimeric virus involves combining the replication complex of the MeV vaccine strain with the envelope glycoproteins (F and H) of wild-type CDV [72]. The resulting recombinant did not cause any clinical signs or immunosuppression in vaccinated ferrets and induced protective immunity from lethal challenge. In addition, several recent attempts employed CDV RGS for the study and the development of CDV vaccines based on wild-type strains (Table 1). One such example is the concept of rational attenuation of the CDV 5804P strain by modifying the L protein so that it expresses EGFP within its second hinge region, leading to viral attenuation in ferrets [73]. Similarly, depleting the *N*-linked glycosylation sites in the H protein of the virus produced a recombinant CDV with an attenuated phenotype, which was no longer able to cause disease in ferrets [74]. 

In general, as a nonsegmented negative-strand RNA virus, the lack of a significant nuclear phase in CDV is a favorable aspect with respect to vector development because it insulates these infectious agents from extensive interference of genetic and cellular processes. Thus, the fact that the CDV genome does not recombine is also of relevance for viral-vector development. In addition, most of CDV vaccine strains showed a good safety record in the context of canines, and diseases arising from Rockborn strain infection such as post-vaccination encephalitis have been stopped by vaccinations [11]. Additionally, the introduction of additional genes or silencing mutations in accessory proteins (C and V) in the viral genome by the RGS could contribute to decreased virulence and immunosuppression of the recombinant CDV in hosts [36,53,73]. Importantly, a virus-based vector must not only be satisfactorily attenuated but must also maintain sufficient replication efficiency in cultured cells for enabling its production in amounts that allow it to be immunogenic in the host. In comparison with other RNA virus vectors, CDV as a replication-competent vector appears to be clearly superior, as it can deliver foreign antigens in vivo with high efficacy for enabling the systemic spread and preferential infection of professional antigen presenting cells (APCs) and lymphoid and epithelial tissues; these vectors also show significant expression efficiency for foreign antigens as candidate multivalent vaccines [31,35,39,69]. 

## 6. CDV as an Oncolytic Virus in Cancer Therapy

Cancer could be defined as the formation of a malignant tumor due to genetic mutations in normal cells leading to their indefinite growth and abnormal proliferation in the body and causing damage to the normal functions of the body. Many cancers remain incurable to modern therapy despite recent antitumor pharmacological regimens in both humans and animals. Roughly 1 to 2% of canines over the age of 10 die from cancer [76,77,78], with the incidence of breast cancer being high, especially in female dogs [79]. Defects in the IFN antiviral response pathway are known to be common in cancer cells, making them more permissive to viral infections compared with their normal counterparts [80]. In humans and the veterinary field, the utilization of morbillivirus, adenovirus, vaccinia virus, coxsackie virus, and reovirus is being extensively investigated and these viruses have entered clinical trials for the treatment of a wide range of advanced cancers [80,81,82,83,84,85,86,87]. These viruses are named oncolytic viruses, as they ideally infect, replicate, and induce the lysis of cancer cells, whereas they do not harm noncancerous cell types. Upon cell lysis, viral progeny are released resulting in the infection of neighboring cancer cells allowing for the easy spread and escalation of the therapy. Oncolytic viruses could be targeted to cancer cells at several levels, including receptor-dependent viral entry, and viral replication [88,89,90]. Recently, increasing studies on oncolytic viruses for cancer treatment have often been focused on engineering the virus to increase viral specificity and enhance the oncolytic effect to cancer cells. In 2015, a new drug for oncolytic virotherapy, talimogene laherparepvec (IMLYGIC), a herpes simplex virus, that has been genetically engineered to express granulocyte macrophage colony stimulating factor (GM-CSF) to treat advanced melanoma was first approved by the Food and Drug Administration (FDA), and was licensed for marketing by the European Commission. 

### 6.1. Mechanisms of Oncolytic Virus and Enhancing Oncolytic Virotherapy Effect through Viral RGS

There are several possible mechanisms through which an oncolytic virus could mediate oncolysis (Figure 3). One of the mechanisms of viral oncolysis is known to be the direct destruction and lysis of tumor cells due to the overwhelming viral replication and release of virions from infected tumor cells, which destruct the tumor vasculature and subsequently trigger cellular defense mechanisms, causing the apoptosis and necrosis of tumor cells [91,92,93] (Figure 3a). Another most likely mechanism is considered to be the induction of cytotoxic immune responses and phagocytosis: the activation of natural killer (NK) cells and CD8+ cytotoxic T-cells triggers lysis of the virus-infected tumor cells, while neutrophils and macrophages exert increased phagocytic activities on tumor cells following viral entry [94,95]. Simultaneously, an increase in the IFN-γ, IL-12, IL-6, and TNF-α proinflammatory cytokines has been reported to foster an accumulation of M1-macrophage derived tumoricidal factors in the tumor microenvironment [96,97] (Figure 3b). In addition, initiation of the complex interplay between IFN-γ, IL-12, NK cells, and the angiotoxic IFN-γ-inducible protein (IP-10) might eventually lead to the depression of tumor angiogenesis and the induction of tumor regression, respectively. As known, IL-12 can mediate the production of IFN-γ by NK cells, and stimulate the accumulation of IP-10. Accordingly, IP-10-stimulated infiltrating NK-cells have been shown to exert a high killing effect on endothelial cells, leading to reduced tumor growth [98,99,100] (Figure 3c). 

Taken together, oncolytic viruses have been shown to prevent neoangiogenesis either by direct infection and destruction of the tumor vasculature, concurrently enhancing innate immunity mediated antitumor effects or by exerting antiangiogenic abilities in tumor tissues. However, before oncolytic viruses can be widely used as therapeutic agents for the clinical treatment of human or animal cancer patients, major obstacles, including potential viral toxicity, ineffective delivery of virus to the tumor, and spread of the virus throughout the tumor mass should first be overcome. To resolve these problems, viral RGS approaches have been recently employed to generate attenuated tumor-selective vaccine strains, as well as selectivity- and efficacy-enhanced, oncolytic virus strains. For instance, MeV was first mentioned as a potential oncolytic agent in anecdotal reports regarding the regression of a Hodgkin lymphoma following natural MeV infection in humans [101]. Attenuated MeV, engineered through RGS to express high affinity single chain T-cell receptors (scTCR), was reported to be retargeted to specific major histocompatibility complex (MHC) ligands [102]. Genetic modifications of the cellular tropism of MeV were achieved by the insertion of tumor-specific ligands at the carboxyl-terminal extensions of the H protein. Single chain antibodies against the carcinoembryonic antigen (CEA), CD20, and CD38 tumor-associated antigens have all been expressed on recombinant MeV strains to facilitate targeted entry to epithelial carcinoma, non-Hodgkin’s lymphoma, and myeloma cells, respectively [103,104,105]. Furthermore, engineering a MeV that could express integrin-binding peptides, cyclic-arginine-glycine-aspartate (RGD) [106], or echistatin [107], has been shown to successfully retarget the virus to endothelial cells of tumor neovessels. To exert the desired therapeutic effects, while reducing its pathogenicity in hosts, a SLAMblind recombinant wild-type MeV strain was rescued and aimed to specifically infect nectin-4 expressing human tumor cells. Results demonstrated that while the SLAMblind MeV maintained an efficient capacity for infection of human breast cancer cells, the virus caused no symptoms, including immunosuppression, in monkeys [108,109]. Interestingly, a further study also demonstrated that the SLAMblind MeV could infect canine mammary cancer cells, displaying antitumor activity in vitro, in xenografts, and ex vivo [86]. Since human and dog nectin-4 proteins share high homology, and the domains critical for the binding of MeV are completely conserved in the two species, this virus could also be considered as a good therapeutic candidate for treating the upregulated expression of nectin-4 in adenocarcinomas of canines [40,110].

### 6.2. CDV as an Oncolytic Virus Displays Broad Cancer Cell Tropism

In the veterinary field, several viruses have been reported as oncolytic agents, including adenovirus, reovirus, vaccinia virus, and CDV. One advantage of CDV as an oncolytic virus is the relatively small size of CDV in comparison with DNA viruses. Only a few proteins of the virus are expressed in addition to those expressed from the artificial inserts, thus minimizing superfluous immune responses in vivo. Second, as members of the morbilliviruses, CDV shares many similarities with MeV, and represents an interesting candidate for its employment in oncolytic therapy, as naturally attenuated vaccine strains have been shown to be able to infect and induce apoptosis in many kinds of cancer cells (Table 2). Infection of canine lymphoid cell lines, including GLGL-90 chronic large granular lymphocytic T-cell leukemia cells, and 17–71 acute B-cell lymphoma cells, expressing SLAM on the surface of their cell membranes with the CDV Onderstepoort strain, has been shown to result in enhanced rates of apoptosis of tumor cells [16]. Similarly, enhanced rates of apoptosis have also been described in a human cervical tumor derived cell line (HeLa cells), after infection with the CDV Lederle strain. Apoptosis of the infected tumor cells was triggered by the intrinsic pathway, as demonstrated by an increase in the amount of the cleaved active form of the caspase-3 protein [18]. In addition, the CDV Onderstepoort strain was able to infect canine histiocytic sarcoma cell lines (CCT and DH82 cells) and induce apoptosis in CCT cells [19]. Persistent infection of DH82 cells with Onderstepoort was shown to lead to increased mRNA transcripts of immune response-activating signal transduction genes, as well as activated M1, and alternatively activated M2 macrophages, whereas it decreased the generation of blood vessels in a xenotransplant mouse model in vivo, investigating the antiangiogenic effect of a CDV infection of tumor cells [19]. Moreover, Onderstepoort-infected DH82 cells were reported to exhibit downregulation of the expression of the matrix metalloproteinase-2 (MMP-2), tissue inhibitors of matrix metalloproteinase-1 (TIMP-1), and TIMP-2, whereas a higher number of mRNA transcripts was observed for the reversion-inducing cysteine-rich protein with Kazal motifs (RECK) [111]. These results suggest that a CDV infection of canine histiocytic sarcoma cells could restore the expression of RECK and might positively influence tumor behavior and reduce its malignant potential [111]. Furthermore, CDV-L strains have also been demonstrated to restrict tumor growth without apparent pathology in a xenograft mouse model and exert antitumor effects in canine mammary tubular adenocarcinoma cells, where they induced apoptosis through the caspase-8 and caspase-3 pathways by activated NF-kB signaling pathway [20]. Recently, it was reported that infection of human adenocarcinoma, breast tumor, and mammary tumor, as well as canine adenofibrosarcoma with the Snyder Hill and Lederle CDV strains led to the death of tumor cells or late apoptosis associated with the expression of TNF-α induced protein 8 (TNFAIP8), strongly suggesting that both human and canine mammary tumor cells could be potential candidates for CDV-induced cancer therapy [17].

### 6.3. Safety Issues Associated with CDV-Based Therapeutics

Safety is of paramount importance. CDV has a non-segmented genome rendering it stable, with a low risk of mutation, thus it is highly unlikely to revert to the pathogenic phenotype. As an effective oncolytic virus for canine cancer cells in vitro, live attenuated CDV strains also maintain the ability to enter the immune and epithelial cells in vivo via the SLAM and nectin-4 receptors. Thus, safety issues with respect to possible virus dissemination and therapeutic effects in animals and humans should be addressed. CDV is thought to be the result of a spillover transmission into dogs from MeV during human epidemics in the New World [3]. The recent notorious host-range expansion of CDV, particularly in non-human primates, and identification of single amino acid changes in the H protein of wild-type strains that allow them to use human SLAM and nectin-4 receptors, highlight the potential of CDV to enter into vacated ecological niches [113]. Although to our knowledge, no report has demonstrated CDV to be a zoonotic pathogen enabling transmission into humans from infected animals, safety issues regarding the use of CDV as an oncolytic virus in humans should also be considered due to the fact that the current MeV vaccine cannot completely protect against CDV infection in monkeys [114]. Moreover, the use of CDV as an oncolytic virus may also raise new biosafety and risk management issues. The risk assessment for trials involving CDV must take into account and mitigate the potential risk of transmission of the infectious agent to wildlife, companion animals, and persons in contact with the treated patients. If necessary, the risk of disease or adverse effects associated with the viral therapeutic could be countered using antiviral agents effective against the CDV strains being considered for cancer treatment.

## 7. Perspectives and Challenges of Oncolytic CDV Therapy

Until now, although there have been few studies on the use of CDV, particularly genetic engineering CDV strains, as an oncolytic virus for canine cancer patients compared with the advanced progress of human virotherapy in clinical trials, the high similarity between human and canine cancers, including colorectal cancer, fibrosarcoma, osteosarcoma, and soft tissue sarcoma [80,81,82,83,84,85,86,87] could help translate cases of successful oncolytic virotherapy from canine to humans and the reverse in a two-way street toward the development of novel antitumor drugs. Despite many similarities between CDV and MeV, including viral tropism, and vaccine-induced host immune responses, both viruses differ significantly in their potential to induce viral invasion of the CNS of hosts, as well as neuropathological lesions via nectin-4-expression-dependent infection in canine brains [41,42]. This difference highlights the fact that CDV as an oncolytic virus could exert the potential ability of neurocytoma virotherapy in canine patients. In addition, CDV infecting a much broader host range, including many carnivore species has been shown to be further opposed to MeV representing a strictly human pathogen. Thus, CDV could not only allow for the extrapolation of possible treatment schemes in veterinary medicine, but also render the canine pathogen an interesting, translational model for various human tumors. The application of oncolytic CDV for human cancers could also avoid the major issue in regard to MeV-based therapeutics: the high prevalence of anti-MeV neutralizing antibodies in the vaccine-immunized population.

Today, several limitations remain in the development of an ideal oncolytic virus, including (1) selective targeting of oncolytic viruses to tumor tissue, (2) relatively poor viral spread throughout solid tumor tissues, (3) inefficient viral replication in immunocompetent hosts, and (4) disadvantageous ratio between the anti-viral and anti-tumoral immunity. As far as CDV is concerned, virotherapeutic studies have been primarily performed using CDV vaccine strains rather than wild-type strains, due to the exceptional genetic stability and safety record in canine populations [16,17,18,19,20]. Nevertheless, most dogs are vaccinated against CDV, and thus the high prevalence of virus-neutralizing antibodies could be one major obstacle restricting the use of CDV as oncolytic viruses and therapeutics for canine cancer patients. Use of genetically engineered viruses through RGS approaches, such as the use of chimeric viruses combining the envelope glycoproteins from MeV or removing key neutralizing epitopes on the surface of viral H or F might help avoid instances of preexisting immunity and overcome such antiviral host immune responses. In addition, intratumoral or mucosal viral application or administration of higher doses of the virus could also be a solution for cases of preexisting immunity problems [115]. However, another problem that needs to be solved to improve its oncolytic effect is the effective and rapid replication and spread of CDV in tumor tissues. In recent studies of two members of the *Mononegavirales,* vesicular stomatitis virus (VSV) and MeV-mediated oncolysis, the strategies used to increase their oncolytic activity employed the use of VSV/MeV hybrid viruses generated by RNA virus RGS as a platform for oncolytic virotherapy [116]. This hybrid incorporated the powerful replication machinery of VSV and encoded both the MeV H and F instead of the VSV-G attachment protein [117]. Accordingly, the VSV/CDV hybrid virus could also be enhanced to spread the cytopathic effect in cancer cells compared with CDV alone. 

Recently, “immunovirotherapy” strategies to enhance the oncolytic efficacy of MeV have been developed with the introduction of immunomodulatory transgenes (GM-CSF, IFN-β, IL-12 and IL-15, etc.) and antibodies against immune checkpoint inhibitors (cytotoxic T-lymphocyte antigen-4, CTLA-4 and programmed cell death-1, PD-1) that stimulate the native anti-tumor immune response. These strategies were used for MeV-susceptible cancer mouse models and showed significantly-senhanced tumor regression or prolonged overall survival, which were correlated with an influx of host neutrophils and tumor-infiltrating cytotoxic T cells [118,119,120,121]. CTLA-4 and PD-1 are inhibitory receptors that limit T cell activation. Tumor cells have been demonstrated to exploit this mechanism of T cell ablation to evade the immune system. Thus, the antibodies blocking CTLA-4 and PD-1 and its ligand PD-L1 show promising anti-tumor effects in a wide range of tumor types by priming T cells against tumor antigens [122,123,124,125]. Taken together, the potential synergistic clinical improvements brought about by employing a combination of immunovirotherapy and clinical treatments, including chemo- and radiotherapy will be vital for future directions in the use of CDV as an oncolytic agent.

## 8. Conclusions

This review summarized the important roles that CDV RGS approaches have played in (1) the construction of recombinant CDV-based vaccines; (2) understanding the mechanisms of the cellular tropism and pathogenesis of CDV from a molecular perspective, comprising not only the interactions of the viral proteins with host cellular receptors but also the influence of host factors on viral virulence; and (3) the potential ability of CDV, particularly vaccine strains, to be used as an oncolytic virus against cancers in humans and animals by targeting tumor markers as viral receptors. However, the action of CDV as an oncolytic virus in human and canine cancers has been primarily evaluated in tumor cells in in vitro systems only. Detailed in vivo studies are highly required for a better understanding of the mechanisms of interactions between the virus and tumor cells, as well as of the impact on the microenvironment and antitumor immune reactions.

## Figures and Tables

**Figure 1 viruses-12-00339-f001:**
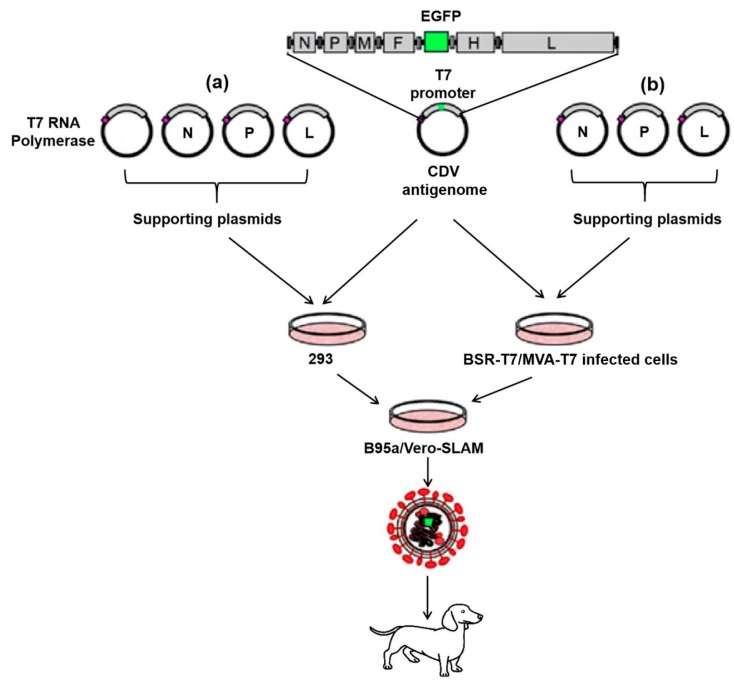
Reverse genetic system for the recovery of a recombinant canine distemper virus (CDV) wild-type strain. The five recombinant plasmids used for the recovery of a recombinant CDV strain, including one T7-promoter driven full-length cDNA expressing plasmid containing the enhanced green fluorescence protein (EGFP) gene inserted between the fusion (F) and hemagglutinin (H) genes, four cytomegalovirus (CMV) promoter-governed supporting plasmids expressing the nucleocapsid (N), phospo (P), and large (L) proteins respectively, and the T7 polymerase plasmid (**a**), which could be replaced by baby hamster kidney cells stably expressing T7 RNA polymerase (BSR-T7) or the modified vaccinia Ankara (MVA-T7) virus (**b**). The recovery process for the generation of the recombinant CDV is as follows: All plasmids are used for the simultaneous cotransfection of 293 cells (**a**), or BSR-T7 cells or MVA-T7 infected cells (**b**), followed by identification of the expression of EGFP in infected foci, and the overlay of B95a or Vero-SLAM (Vero cells expressing the signaling lymphocytic activation molecule [SLAM] receptor) cells for subsequent viral propagation.

**Figure 2 viruses-12-00339-f002:**
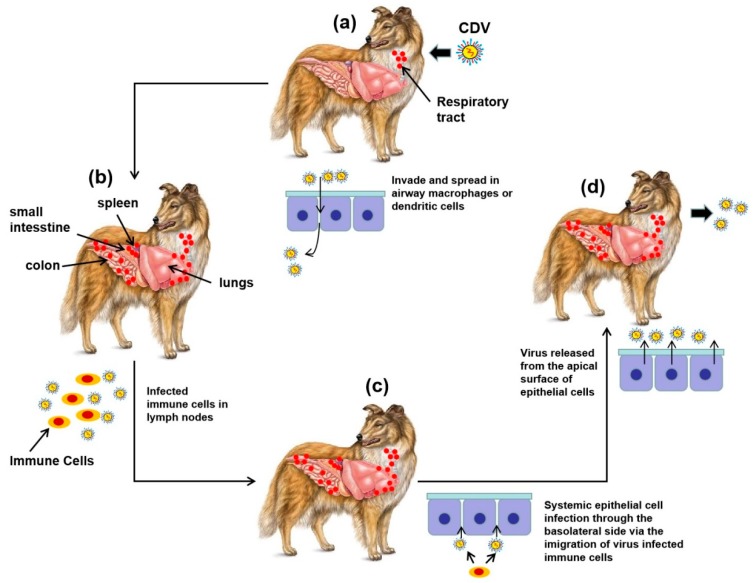
Schematic of the entry cells and tissues, dissemination, and shedding process of CDV in infected dogs. (**a**) CDV infects macrophages or dendritic cells of the respiratory epithelium by binding with SLAM; (**b**) immune cells of the lymph nodes are infected and then the CDV disseminates to the spleen, thymus, etc.; (**c**) CDV further infects epithelial tissues by binding to the receptor nectin-4 of epithelial cells of the basolateral surface; (**d**) CDV is released from the apical surface of epithelial cells.

**Figure 3 viruses-12-00339-f003:**
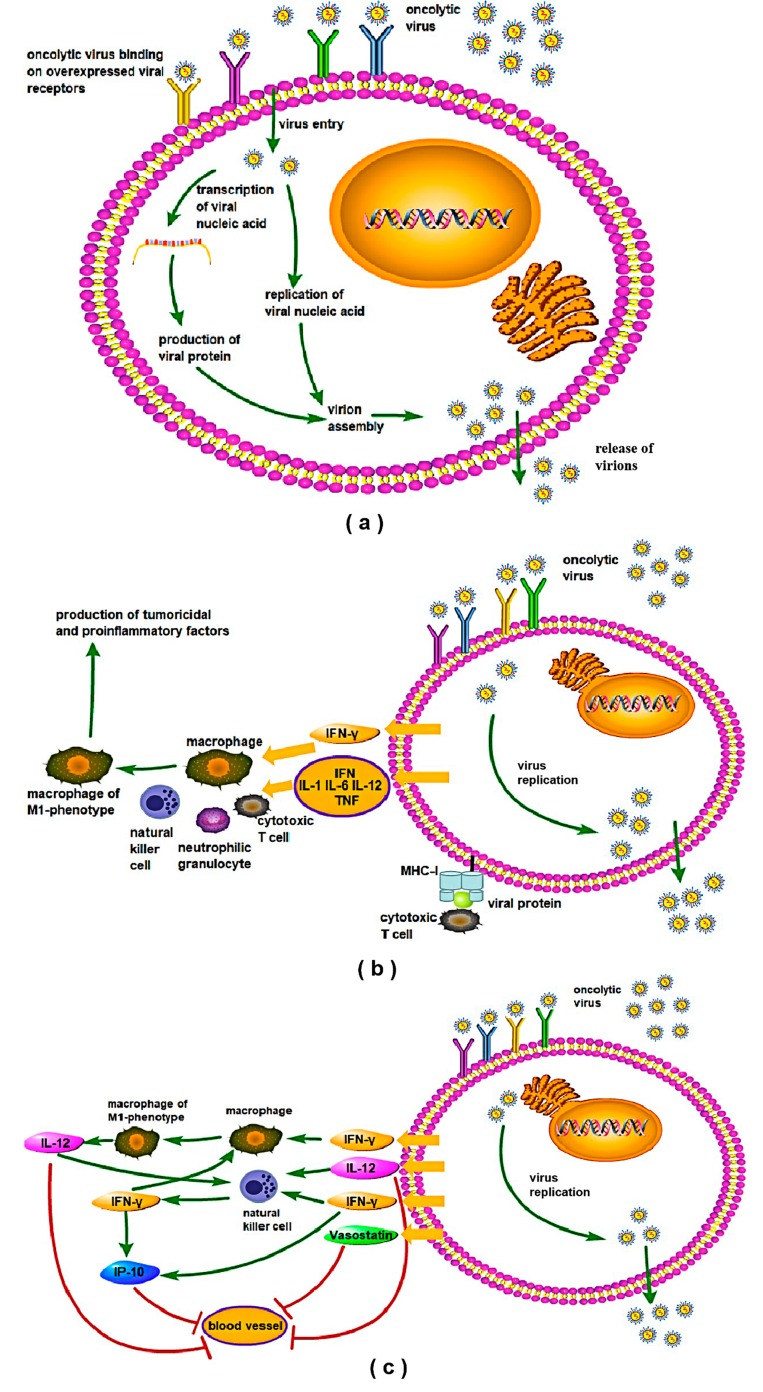
Possible mechanisms of the lysis of tumor cells upon infection with an oncolytic virus. (**a**) Binding to receptors initiates the internalization of the virus, which then release the viral nucleic acid, starting viral proliferation and release, leading to the apoptosis of tumor cells. (**b**) A pathogen-associated molecular pattern stimulates the production of antiviral cytokines (IFN-γ, IL-1, IL-6, IL-12, tumor necrosis factor [TNF]), which leads to the activation of immune cells, mediating cytotoxicity and phagocytosis. Meanwhile, major histocompatibility complex (MHC)-I-mediated presentation of viral proteins activates CD8+ cytotoxic T-cells, triggering lysis of the oncolytic virally-infected tumor cells. (**c**) Secretion of IFN-γ and IL-12 might initiate the complex interplay between IFN-γ, IL-12, natural killer (NK)-cells, and IFN-γ-inducible protein (IP-10), leading to the eventual limitation of tumor angiogenesis. Moreover, externalization of vasostatin is also involved in the inhibition of tumor angiogenesis. The arrows indicate the various steps and the components of an oncolytic virus during exerting the lysis of tumor cells, with the bar indicating their participation inhabiting the tumor angiogenesis.

**Table 1 viruses-12-00339-t001:** Recombinant CDV generated using CDV reverse genetic system (RGS).

CDV	Virus Strain	Foreign Antigen or Reporter Gene Expressed	Genomic Position	Immune Responses or Virulence	Ref.
Attenuated strain	Onderstepoort	Luciferase	P-M	NFAE	[75]
	R-20/8	RABV-G	P-M	Protection of mice from RABV challenge and introduction of dogs with NAb against RABV and CDV	[35]
	CDV-L	RABV-G	N-P	Introduction of mice with specific NAb against both RABV and CDV	[31]
	Yanaka	EGFP/Luciferase	N-P	NFAE	[37]
	Yanaka	IL-18	N-P	Introduction of increased IFN-γ expression in PBMCs and splenocytes of dogs	[70]
	Yanaka	*Leishmania* Antigen	N-P	Protection of dogs from challenge with CDV and cutaneous leishmaniasis	[69]
Wild-type strain	A75/17-V	EGFP	3′ Leader-N	NFAE	[29]
	5804P	EGFP	3′ Leader-N, P-M, H-L	Virulence in ferrets depended on the position of EGFP insertion	[36]
	5804P	EGFP	within L	Overattenuation and protection of ferrets against challenge with the virulent parental virus	[73]
	SnyderHill	EGFP/dTomato	H-L	Introduction of ferrets with CNS disease without loss of virulence	[38]
	SnyderHill	Venus/dTomato/TagBFP	H-L	Introduction of ferrets with CNS disease without loss of virulence	[48]
	Wuhan-15	IL-7	P-M	Facilitating the generation of follicle helper T or germinal center B cells in mice and enhancing the production of CDV NAb	[71]

RABV-G: rabies virus G protein; NFAE: no foreign antigen expressed; NAb: neutralizing antibodies; IFN: interferon; PMBCs: peripheral blood mononuclear cells; EGFP: enhanced green fluorescence protein; IL: interleukin; CNS: central nervous system; Ref.: references.

**Table 2 viruses-12-00339-t002:** Summary of CDV strains used for oncolysis.

Virus Strain	Tumor Type	Tumor Cells	Oncolytic Effects In Vivo	Ref.
Onderstepoort	Canine lymphoma	CLL-1390, CLGL-90, 17–71	ND	[16]
Lederle	Human cervical tumor	HeLa	ND	[18]
Onderstepoort	Canine histiocytic sarcoma	CCT, DH82	Decreasing blood vessel generation with a lower blood vessel density in an in vivo mouse model	[19] [111] [112]
Snyder Hill, Lederle	Human adenocarcinoma, Human breast tumor, Human mammary tumor, Canine denofibrosarcoma	C2Bbel, HS578T, MCF-7, AF-72	ND	[17]
CDV-L	Canine mammary tubular adenocarcinoma	CIPp	Restriction of tumor growth without any apparent pathology in a xenograft mouse model	[20]

ND: not done; Ref.: references.

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
