# Peer review of "Viral Pathogenesis, Recombinant Vaccines, and Oncolytic Virotherapy: Applications of the Canine Distemper Virus Reverse Genetics System"

_viruses, 2020, doi:10.3390/v12030339_

Round 1

Reviewer 1 Report

This is a review manuscript which has a potential. My concerns are first related to the merging of important facts, vaccine sand oncolytic power.

The authors should better explain the rationale behind this.

I would change the tithe "Lights and Shades on CDV Vaccines", is has been overused in the recent years regarding vaccines.

Author Response

Point 1: This is a review manuscript which has a potential. My concerns are first related to the merging of important facts, vaccines and oncolytic power. The authors should better explain the rationale behind this.

Response 1: In the revised manuscript, we supplied and reviewed the details of immune responses triggered by the CDV-based vaccines (red column in Table 1) and CDV oncolytic power in cancer animal model (red column in Table 2). Additionally, the rationales of effects on vaccines and oncolytic virotherapy have been also made a supplement in revised version according to the comments of reviewer (Page 7-8: line 280-296; Page 13-14: 481-494).

Point 2: I would change the tithe "Lights and Shades on CDV Vaccines", is has been overused in the recent years regarding vaccines.

Response 2: Yes, I agree with you! We change the title "Lights and Shades on CDV Vaccines" to “History and Challenges associated with CDV Vaccines”. Now, it looks like the topic and content in this paragraph is matching in the review. (Page 6: line 217)

Reviewer 2 Report

The review article entitled "Recombinant Canine Distemper Virus-based Vaccines and Oncolytic Virotherapy" is well written and well structured. The review covers important aspects of canine distemper virus (CDV) in relation to its potential anti-cancer impact in veterinary medicine.

However, several points should be addressed before publication:

  • The title refers to the usage of attenuated CDV as a potential platform to develop oncolytic vectors (OVs) as well as next-generation vaccines. In the manuscript, however, the authors are not developing much the opportunity to employ reverse genetics systems (RGS) for the engineering of new CDV-based vaccines. The review almost exclusively focuses on the potential to use CDV as an oncolytic vector. For instance, and considering the safety records of the CDV vaccine strain, the latter may offer an attractive basis for the development of multivalent vaccines. The authors should address this point.
  • The paragraph describing RGS is not mentioning latest improvements in the fields. For instance, the authors may describe the usage of an optimized T7 promoter together with the hammerhead ribozyme sequence in front of the CDV leader.
  • The authors may also discuss safety issues in correlation to the usage of OVs in general. Regarding this important factor, performing attenuated CDV-based clinical trials in a country where CDV vaccination is mandatory, may offer an additional level of confidence to ethical committees.
  • The authors are mentioning to employ CDV attenuated strains (and eventually derived engineered variants) to treat human tumors. Again, the authors should be careful in mentioning this with regard to safety issues and realistic clinical investigations.
  • As mentioned by the authors, oncolytic virotherapy platforms do show drawbacks, such as inefficient intratumroal viral spread and/or limited cytolytic activity and rapid clearance of the OV by the immune system. Although this review focuses on OVs, a paragraph briefly describing immune checkpoint inhibitors and related drawbacks (e.g. lack of pre-infiltrated cytotoxic T cells) might be very interesting for the authors. This may allow speculating on potential synergistic clinical improvements by employing combined treatments.

Author Response

Point 1: The review article entitled "Recombinant Canine Distemper Virus-based Vaccines and Oncolytic Virotherapy" is well written and well structured. The review covers important aspects of canine distemper virus (CDV) in relation to its potential anti-cancer impact in veterinary medicine. However, several points should be addressed before publication:

Response 1: We appreciate the positive opinions of the reviewer. According to the review’s comments, all the points in the manuscript should be addressed has been revised and shown below.

Point 2: The title refers to the usage of attenuated CDV as a potential platform to develop oncolytic vectors (OVs) as well as next-generation vaccines. In the manuscript, however, the authors are not developing much the opportunity to employ reverse genetics systems (RGS) for the engineering of new CDV-based vaccines. The review almost exclusively focuses on the potential to use CDV as an oncolytic vector. For instance, and considering the safety records of the CDV vaccine strain, the latter may offer an attractive basis for the development of multivalent vaccines. The authors should address this point.

Response 2: First, we revised the title with “Viral Pathogenesis, Recombinant Vaccines, and Oncolytic Virotherapy: Application of the Canine Distemper Virus Reverse Genetics System”. So it looks like the title and content of the review is matching (Page 1, line 2-4). Second,  we have developed more opportunity to employ RGS for engineering of CDV-based vaccines (Page 3: line 93-98, line 106-115; Page 7: line 245-251, line 267-279). Finally, we addressad the development of multivalent vaccines considering the safety records of CDV vaccine strain according to the review’s comments (Page 7-8: line 280-288).

Point 3: The paragraph describing RGS is not mentioning latest improvements in the fields. For instance, the authors may describe the usage of an optimized T7 promoter together with the hammerhead ribozyme sequence in front of the CDV leader.

Response 3: Thank you for the nice reminder! That is very useful to our works in future. In the revised version, we have described the latest RGS improvements and cited the related references, which described the usage of an optimized T7 promoter together with the hammerhead ribozyme sequence in front of the viral 5’-end antigenome. These improved RGS have substantially increased rescue efficiency for representative viruses from all 5 major Paramyxoviridae genera (Page 3: line 106-115).  

Point 4: The authors may also discuss safety issues in correlation to the usage of OVs in general. Regarding this important factor, performing attenuated CDV-based clinical trials in a country where CDV vaccination is mandatory, may offer an additional level of confidence to ethical committees.

Response 4: Yes, I think it is necessary to let the readers to know the safety issue in correlation to the usage of oncolytic viruses. In the revised manuscript, we not only reviewed the safety issue of the negative-strand RNA virus based vaccines, particularly CDV (Page 7-8: line 280-294), but also e focused on the safety issue of application of CDV-based therapeutics for animals and humans. (Page 12-13: line 420-440)

Point 5: The authors are mentioning to employ CDV attenuated strains (and eventually derived engineered variants) to treat human tumors. Again, the authors should be careful in mentioning this with regard to safety issues and realistic clinical investigations.

Response 5: Yes! The potential pathogenic risks in humans and biosafety issues of attenuated CDV or derived engineered variants in virotherapy for treating human tumours were discussed in the revised manuscript. In addition, the adverse effect related to realistic clinical applications was also including. (Page 12-13: line 420-440)

Point 6: As mentioned by the authors, oncolytic virotherapy platforms do show drawbacks, such as inefficient intratumroal viral spread and/or limited cytolytic activity and rapid clearance of the OV by the immune system. Although this review focuses on OVs, a paragraph briefly describing immune checkpoint inhibitors and related drawbacks (e.g. lack of pre-infiltrated cytotoxic T cells) might be very interesting for the authors. This may allow speculating on potential synergistic clinical improvements by employing combined treatments.

Response 6: Yes, we are also interested in the immune checkpoint inhibitors for the anti-tumour therapy. In the revised manuscript, the recent development of “immunovirotherapy” including the strategies immune checkpoint inhibitors (antibodies blocking CTLA-4 and PD-1 and its ligand PD-L1) was described. Thus, the potential synergistic clinical improvements by employing combined immunovirotherapy and clinical treatments, including chemo- and radiotherapy will be vital for future directions in the use of CDV as an oncolytic agent.(Page 13-14: line 481-494)